# Is the Straight Leg Raise Suitable for the Diagnosis of Radiculopathy? Analysis of Diagnostic Accuracy in a Phase III Study

**DOI:** 10.3390/healthcare11243138

**Published:** 2023-12-11

**Authors:** Alberto Montaner-Cuello, Elena Bueno-Gracia, Diego Rodríguez-Mena, Elena Estébanez-de-Miguel, Miguel Malo-Urriés, Gianluca Ciuffreda, Santos Caudevilla-Polo

**Affiliations:** Physiatry and Nursery Department, Health Sciences Faculty, University of Zaragoza, 50009 Zaragoza, Spain; albertomontaner@unizar.es (A.M.-C.); drodriguezm@salud.aragon.es (D.R.-M.); elesteba@unizar.es (E.E.-d.-M.); malom@unizar.es (M.M.-U.); gciuffreda@unizar.es (G.C.); scp@unizar.es (S.C.-P.)

**Keywords:** diagnosis, mechanosensitivity, lumbosacral radiculopathy, straight leg raise

## Abstract

The straight leg raise test (SLR) has been proposed to detect increased nerve mechanosensitivity of the lower limbs in individuals with low back pain. However, its validity in the diagnosis of lumbosacral radiculopathy shows very variable results. The aim of this study was to analyse the diagnostic validity of the SLR including well-defined diagnostic criteria (a change in symptoms with the structural differentiation manoeuvre and the reproduction of the patient’s symptoms during the test or the asymmetries in the range of motion or symptoms location between limbs) in a sample of participants in phase III with suspicion of lumbar radiculopathy using the electrodiagnostic studies (EDX) as the reference standard. A phase III diagnostic accuracy study was designed. In total, 142 individuals with suspected lumbosacral radiculopathy referred for EDX participated in the study. Each participant was tested with EDX and SLR. SLR was considered positive using three diagnostic criteria. The sensitivity of the SLR for Criterion 3 was 89.02% (CI 81.65–96.40), the specificity was 25.00% (CI 13.21–36.79), and the positive and negative likelihood ratios were 1.19 (CI 1.01–1.40) and 0.44 (0.21–0.94), respectively. SLR showed limited validity in the diagnosis of lumbosacral radiculopathy. The incorporation of more objective diagnostic criteria (asymmetry in range of motion or localisation of symptoms) improved the diagnostic validity but the imprecision of the confidence intervals limited the interpretation of the results.

## 1. Introduction

Radiculopathy is defined as a dysfunction of a spinal nerve root that can cause pain, weakness, sensory alterations, and/or decreased myotatic reflexes in a specific anatomical territory corresponding to the level of the affected root [1,2,3]. Lumbosacral radiculopathy or lumbosacral radicular syndrome refers to dysfunctions of the roots that correspond to the formation of the lumbar and/or sacral spinal nerves [4]. Although it is difficult to establish specific figures, some authors calculate a prevalence of between 1–5% of the general population, affecting men and women equally [5,6]. In any case, the costs derived from the direct and indirect treatment of lumbosacral radiculopathies can be very high [7].

The aetiology of lumbosacral radiculopathy is usually related to phenomena of mechanical and/or chemical origin [8,9,10]. Mechanical injury to the nerve root can occur by compression, traction, or friction forces. Chemical irritation may occur in response to nerve root ischemia, vascular stasis, or exposure of the root to inflammatory components released during tissue injury [8,11,12]. Compressive aetiology caused by disc herniation and/or stenosis of the foramina is the main known and studied cause of lumbosacral radiculopathy [4,13,14,15].

The diagnosis of radiculopathy requires correlation of the results of different components of the evaluation process such as: anamnesis, assessment of sensation and strength, diagnostic imaging, and electrodiagnostic studies [5,11]. Each of these components of the evaluation provides unique and relevant information which helps to confirm or reject diagnostic suspicions [5,10,11,16]. Traditionally, magnetic resonance imaging (MRI) has been used to diagnose lumbosacral radiculopathy due to lumbar disc herniation or other degenerative processes of the lumbosacral complex [15,17]. However, the aetiology of lumbar radiculopathy may include causes other than lumbar disc herniation. Therefore, electrodiagnostic studies (EDX) are used to confirm the diagnosis of lumbosacral radiculopathy [11,18].

Neurodynamic tests (NDTs) are physical tests used to diagnose lumbosacral radiculopathy. NDTs consist of a sequence of movements that selectively increase the neural structures to be evaluated, producing a mechanical stimulus when they are subjected to sufficient tension [19,20,21,22,23]. The straight leg raise test (SLR) and the slump test are the two most used NDTs in the diagnosis of lumbosacral radiculopathy. Although the SLR has been proposed to detect increased nerve mechanosensitivity of the lower limbs in individuals with low back pain [10,23,24,25], previous studies on its validity in the diagnosis of lumbosacral radiculopathy show very variable results [10,26,27]. The variable validity of the test could be related to several aspects. First, the interpretation of the tests—although authors such as Nee et al. [28] proposed that at least two premises should be fulfilled to consider a test positive (abnormal) to make the diagnostic criteria more clearly defined: (1) reproduction of the patient’s symptoms during the NDTs, and (2) a change in those symptoms by structural differentiation (SD)—there is a lack of specificity in many studies and there is no unification of the diagnostic criteria. The fact that the SLR results are usually compared with the observation of lumbar disc herniation in MRI [15,17,29] is another possible reason for the poor validity results.

Considering these aspects, the alternative hypothesis of this study was that the SLR is a valid NDT to diagnose lumbosacral radiculopathy if well-defined diagnostic criteria (i.e., reproduction of the patient’s symptoms during the test and or the asymmetries in the range of motion or symptoms location between limbs) are used for what is considered a positive test and a change in symptoms with the structural differentiation manoeuvre is a fundamental criterion to consider the SLR as positive. Therefore, the aim of this study was to analyse the diagnostic validity of the SLR with well-defined diagnostic criteria in a sample of participants in phase III with suspicion of lumbar radiculopathy. The EDX was used as a reference standard to include radiculopathies caused by a different aetiology than lumbar disc herniation.

## 2. Methodology

### 2.1. Study Design

A phase III diagnostic accuracy study was designed following the classic classification of Sackett and Haynes [30]. Phase III diagnostic accuracy studies aim to determine if the diagnostic test under study distinguishes the subjects with and without the pathology among those who are suspected of having the pathology in an everyday clinical situation and are subjected to the reference diagnostic test [30]. The study followed the 2015 STARD reporting standards [31]. The SLR was used as the index test and the EDX as the reference standard for the diagnosis of LSR. To calculate the sample size, the tables elaborated by Hajian-Tilaki were used [32]. Based on the previous data from the Clinical Neurophysiology Service of the study, a prevalence of 50% of the pathology in the study population was taken into account. Based on a confidence interval of 95%, a margin of error of 10%, and a sensitivity and specificity between 75% and 80%, a sample size range of between 123–161 subjects was required. The study was conducted in accordance with the ethical principles and the Helsinki Declaration on research involving human subjects. The Ethical Committee for Clinical Research of Aragon (CEICA) approved the protocol of this study (PI21/073).

### 2.2. Participants

Participants were recruited from consecutive patients with suspected lumbosacral radiculopathy referred to the Clinical Neurophysiologist Department of the Clinical University Hospital “Lozano Blesa” in Zaragoza (Spain) and were invited to voluntarily participate in the study. Patients were informed about the study and they gave their consent for participation before inclusion. Inclusion criteria were: patients aged up to 18 years old [33] presenting symptoms compatible with lumbosacral radiculopathy for more than 3 weeks at the time of the study—intermittent or constant pain in the lumbar area and radiating to a distal extremity, to the gluteal fold; or distribution of pain according to a dermatomic pattern or weakness according to a myotonic pattern of L4, L5, and/or S1 nerve roots—[34,35] and having sufficient understanding and communicative capacity to communicate their symptoms, as well as their characteristics [36]. The main exclusion criteria for participation were: any diagnosis of diabetes mellitus type I, unregulated thyroid dysfunctions, rheumatoid arthritis, serious heart and/or lung diseases, alcoholism, HIV+, herpes zoster infection, multiple sclerosis, hereditary neuropathy, known pregnancy and/or serious systemic or autoimmune diseases [34,36,37,38,39], having undergone surgery or suffered fractures in the lumbar spine in the last year [36], any ROM limitation of the lower limbs that prevented SLR testing [37], inability to lie supine [40], and any physical contraindications for physical therapy [34,37,38,41].

### 2.3. Reference Standard

EDX included electromyography (EMG) and electroneurography (ENG). EMG was performed with a concentric needle electrode. The “H” reflex was recorded, and ENG was used to exclude other diseases and confirm the diagnosis. The algorithm established by the American Association of Electrodiagnostic and Neuromuscular Medicine (AANEM) was followed to perform the EDX—followed by the main manuals [42]—which consists of exploring the innervated muscles at the segmental level that corresponds to the suspicion of radiculopathy. Additional muscles with the same segmental innervation were studied to confirm the diagnosis if one of the muscles examined showed abnormalities. Additionally, the study was completed with ENG to find out if this abnormality was due to mononeuropathy. If none of the muscles showed an abnormal examination, radiculopathy was ruled out [43].

### 2.4. Index Test

Patients were informed of the possibility of participating in the study after performing EDX, and those who met the inclusion criteria, agreed to participate in the study, and signed the informed consent were included. Demographic variables—age, gender, and body mass index (BMI)—variables related to the patient’s clinical presentation—location and duration of symptoms—and the ND4 and Oswestry [44,45] questionnaires were recorded. The SLR was performed approximately 30 min after the EDX by a single physiotherapist with more than 10 years of experience in neurodynamics. The evaluator was blinded to the EDX results.

The SLR was performed as follows [23]: participants were positioned supine, their arms alongside their bodies, and lower limbs straight. Then, hip flexion was performed with the knee extended until the patient reported the first appearance of symptoms (P1). At that point, a structural differentiation manoeuvre was performed. The structural differentiation consisted of dorsiflexion of the ankle when the patient reported proximal symptoms (above the popliteal fossa), and when the symptoms were distal (below the popliteal fossa) the structural differentiation consisted of reducing a few degrees of hip flexion. If the symptoms changed with structural differentiation, they were classified as of neural origin and if they did not change, as musculoskeletal.

Once the SLR and the structural differentiation manoeuvre were performed, the range of movement (ROM) at P1 and the distribution of the sensory response were recorded. The ROM was measured with the “angle meter” application, version 1.3.0. This measurement with digital inclinometers has shown excellent validity and reliability for measuring hip flexion and, specifically, the SLR test [46]. The device was placed using a cuff on the patient’s tibial shaft 10 cm from the peroneal malleolus. The starting point was considered 0 degrees with the patient in the supine position and the degrees were recorded at the onset of the patient’s symptoms. Participants were also asked to indicate the the quality and distribution of the sensory responses. A body chart was used and participants were asked to mark the location of the perceived sensory responses during the SLR. Subsequently, data were extracted from the body charts according to 6 areas: lumbar, pelvis, thigh, popliteal fossa, calf, and foot. To describe the quality of the perceived symptoms, participants had to choose between the following descriptors: pain, stretching, tingling, pricking, numbness, burning, or other.

The diagnostic validity of the SLR was analysed for 3 different diagnostic criteria of the SLR:

Criterion 1: reproduction of the patient’s clinical symptoms <70° of hip flexion;

Criterion 2: reproduction of the patient’s clinical symptoms <70° of hip flexion and change in symptoms with the structural differentiation manoeuvre;

Criterion 3: reproduction of the patient’s clinical symptoms and change in symptoms with the structural differentiation manoeuvre OR change in symptoms with structural differentiation and asymmetry between affected and non-affected limbs (asymmetry ≥ 10° in the hip ROM and/or asymmetry in symptoms location).

### 2.5. Statistical Analysis

The statistical analysis of all variables was performed with the IBM^®^ SPSS^®^ Statistics 21. The confidence level established for the analysis of the results and statistical inference was 95%. The alternative hypothesis was accepted as true with a margin of error of 5%, that is, *p* < 0.05.

A descriptive analysis of the demographic variables and the variables that characterised the symptoms was carried out. For the descriptive analysis of the quantitative variables, the descriptive central tendency of the mean with the standard deviation was used. The distribution of the quantitative variables was analysed using the Kolmogorov–Smirnov statistic with the Lillierfors correction to describe the normal distribution or not. For qualitative variables, frequency and percentages were calculated [47].

A two-by-two contingency table for SLR results and lumbosacral radiculopathy diagnosis was developed. Sensitivity, specificity, validity index, predictive values, and prevalence were calculated [48]. An arbitrary cut-off point was set at 0.75 for sensitivity and specificity (less than 0.60 was considered low diagnostic performance, 0.60 to 0.75 was considered moderate, 0.75 to 0.85 a good value and above 0.85 an excellent value) [49,50]. Likelihood ratios (LR) were also calculated. The +LR was calculated as sensitivity/(1 − specificity) and the −LR was calculated as (1 − sensitivity)/specificity [51]. The diagnostic accuracy of the SLR was considered satisfactory with +LR > 2 or −LR < 0.50 [52].

Finally, the results of the SLR were analysed with different diagnostic criteria by creating flow diagrams of the diagnostic tree of each test, calculating the significance of each decision node with the chi-square statistical test, and analysing how the different stratifications of the diagram behave [53].

## 3. Results

### 3.1. Sample Characteristics

After applying exclusion criteria, 142 participants were enrolled in the study, Figure 1 shows the flow chart of the study according to the Standards for Reporting of Diagnostic Accuracy [31]. The sample was made up of 82 women (57.7%) and the average age of the sample was 54.82 ± 12.33 years. The demographic and symptom characteristics of the sample are shown in Table 1. Eighty-two participants (57.7% of the total) obtained a diagnosis of lumbosacral radiculopathy based on EDX.

### 3.2. SLR Results

In participants diagnosed with lumbosacral radiculopathy, the SLR produced a neurodynamic response in 93.9% of the participants and 70.7% reported a reproduction of their symptoms during the test. The mean ROM at P1 was 39.72 ± 16.3 degrees of hip flexion in the affected limb and 53.30 ± 13.83 degrees in the non-affected limb. The main symptom reported by the patients was tightness (76.8%) and the areas where symptoms were most manifested were the calf (17.1%), the popliteal fossa (14.6%), and the back of the thigh (12.2%). Finally, in 75.6% of the cases the SLR showed asymmetry in the symptom area and in 60.2% asymmetry in the ROM. Table 2 shows the results for the SLR variables, differentiating cases diagnosed with lumbosacral radiculopathy by EDX and cases without radiculopathy.

The SLR was considered positive in 105 participants (73.9%) applying Diagnostic Criterion 1, in 100 participants (70.4%) applying Diagnostic Criterion 2, and in 118 participants (83.1%) applying Diagnostic Criterion 3. Criterion 1 showed a sensitivity of 75.61% (CI 65.71–85.51) and a specificity of 28.33% (CI 16.10–40.57), with a +LR of 1.06 (CI 0.86–1.29) and a −LR of 0.86 (CI 0.49–1.50). Criterion 2 showed a sensitivity of 73.17% (CI 62.97–83.37) and a specificity of 33.33% (20.57–46.09), with a +LR of 1.10 (CI 0.88–1.37) and a −LR of 0.80 (0.49–1.33). Finally, Criterion 3 showed a sensitivity of 89.02% (CI 81.65–96.40) and a specificity of 25.00% (CI 13.21–36.79), with a +LR of 1.19 (CI 1.01–1.40) and a −LR of 0.44 (CI 0.21–0.94). Table 3 shows the results of the SLR validity for the three diagnostic criteria, and Figure 2 represents the flow charts for each of the diagnostic criteria.

## 4. Discussion

The present study analysed the validity of three different diagnostic criteria of positive SLR in the diagnosis of lumbosacral radiculopathy. Three different diagnostic criteria were used to compare the results in the same sample, since there is a great disparity in criteria in previous studies in different samples. Criterion 1 was the classic interpretation criterion of the SLR test, in which the reproduction of the patient’s symptoms was considered positive and in a range <70° of hip flexion [22]. In Criterion 2, the change in the patient’s symptoms with the structural differentiation manoeuvre was included as a fundamental criterion to consider the test positive, as recommended by many authors [23,28]. Finally, in Criterion 3, in addition to the reproduction of symptoms with structural differentiation, any asymmetry was added, both in the ROM—asymmetry >10° in hip flexion—and in the location of the symptoms, with structural differentiation even if it did not reproduce symptoms in order to consider the test as positive [23,28,35]. The aim of Criterion 3 was to increase the specificity of the test.

Although it is true that the sensitivity value of the test, as well as the +LR and −LR values, were better for Criterion 3, none of the three criteria for interpreting the SLR showed a good validity in the diagnosis of lumbosacral radiculopathy. Although the test showed poor results for the characteristics of the study carried out, Criterion 3 showed more diagnostic precision than the rest of the criteria. Criterion 3 was the only criterion where the chi-square test resulted significant. Specifically, based on the −LR results, Criterion 3 reflected some utility in modifying the diagnosis of lumbosacral radiculopathy in the event of a negative result in the SLR. Specifically, the clinical interpretation of the results obtained would be that, if the SLR test does not reproduce the patient’s symptoms and there are no asymmetries in the ROM or symptoms location even though the structural differentiation has produced a change in the symptoms, the probability that the patient does not have lumbosacral radiculopathy is high. However, imprecision in the confidence intervals (CIs) and the high number of false positives limit interpretation from the data and it should be concluded that the test alone cannot be used to diagnose lumbosacral radiculopathy.

The low validity for the SLR test obtained in the present study, compared to other studies, could be related to several factors. Firstly, the reference test used. Most systematic reviews and studies use MRI as a reference standard [10,17] and it has been proven that MRI can diagnose a positive in asymptomatic people [54]. This factor may have overestimated the diagnostic accuracy of SLR. Using EDX as a reference, which has a low false positive rate, could provide values closer to reality. Furthermore, these studies aim to find the herniated disc and not the radiculopathy [39], when they may be two different concepts.

Secondly, the type of sample analysed. In some studies the sample is obtained from subjects who have already tested positive in the reference standard or who have had mixed positive and negative results previously [33,35,37,55] but the present study was performed in a situation of diagnostic uncertainty (phase III). It is essential to keep in mind that TNDs must be performed on a sample obtained from a study population exactly the same as that on which the EDX and imaging tests are performed. Studies that perform the test on a sample with confirmed pathology and compare results to healthy subjects or a sample consisting of subjects who have already tested positive in the standard test are not methodologically correct to compare at this point. In phase III, the characteristics of the sample are different from those of studies in phase I and II, since they are patients with a long evolution, with a high probability of having associated pathologies and being more sensitised. This aspect makes it more difficult to differentiate between subjects with radiculopathy and other pathologies. In fact, some authors claim that NDTs are good tests for detecting changes in neural mechanosensitivity [28], rather than for detecting a specific pathology. Probably, many of the patients in the sample had some other pathology (sometimes neural) different from lumbosacral radiculopathy. This could explain why the SLR has obtained many false positives for lumbosacral radiculopathy.

Other possible explanations for the low validity of the SLR in the diagnosis of radiculopathy could be related to more intrinsic factors of the test. For example, the SLR test does not pre-tension the nerve roots or the spinal cord as the slump test does [23] and that could influence the mechanosensitive response. In the results of the present study, many positive symptom reproduction responses may have been due to a conjunction of musculoskeletal pain together with the neural response of the sciatic nerve and not so centred in the nerve root. The positive structural differentiation would allude to this decrease in symptoms at the level of the sciatic response, but not the pain caused in the musculoskeletal structures of the lumbar area. Also, the entire posterior muscular package is extended and it is easier to combine the neural and muscular response. Coupled with the previous reasoning, the SLR is performed with hip flexion and is an anatomical area close to the lumbosacral and iliac region, the source of the pain. Initially moving that anatomical area to perform the test puts stress on many structures that are suffering from chronic pain due to various mechanisms because surely our patient in phase III has, in addition to a possible radiculopathy, radiated myofascial trigger point pain, facet joint syndrome, discogenic pain, or vertebrogenic pain.

### Study Limitations

This study has several limitations. Firstly, NDTs include an element of subjectivity in the diagnosis since the classification of a finding as positive or negative resides in the reproduction of the patient’s symptoms. Also, the effect of structural differentiation must be indicated by patients. To try to minimise subjectivity, Criterion 3 was proposed in the study, adding asymmetry in the ROM or in the location of symptoms as possibly more objective aspects to make the diagnosis [23,28]. Another important aspect was the reference test to be compared with the SLR results. We know that EDX has a very low false-positive rate, but we would have liked to be able to compare EDX results with MRI as a complement, even though we know that this test has a positive rate in an asymptomatic population. We discarded the observation in surgical intervention because it greatly biases the study sample. Another potential limitation of the study is related to the CIs. Although the -LR results obtained for Criterion 3 could indicate the ability of the SLR to generate small shifts from pre-test to post-test probability, imprecision in the CIs limits interpretation from the data. Finally, the SLR was performed after the EDX. Although a 30 min break was respected between EDX and SLR, the fact that participants were tested after the EDX may have altered the validity of the nerve roots and potentially biased the results.

## 5. Conclusions

The results of the study indicate that the SLR has limited validity in the diagnosis of lumbosacral radiculopathy, using the EDX as a reference standard and in a phase III sample. The incorporation of more objective diagnostic criteria (asymmetry in the ROM or localisation of symptoms) improves diagnostic validity but the imprecision of the confidence intervals limits the interpretation of the results.

## Figures and Tables

**Figure 1 healthcare-11-03138-f001:**
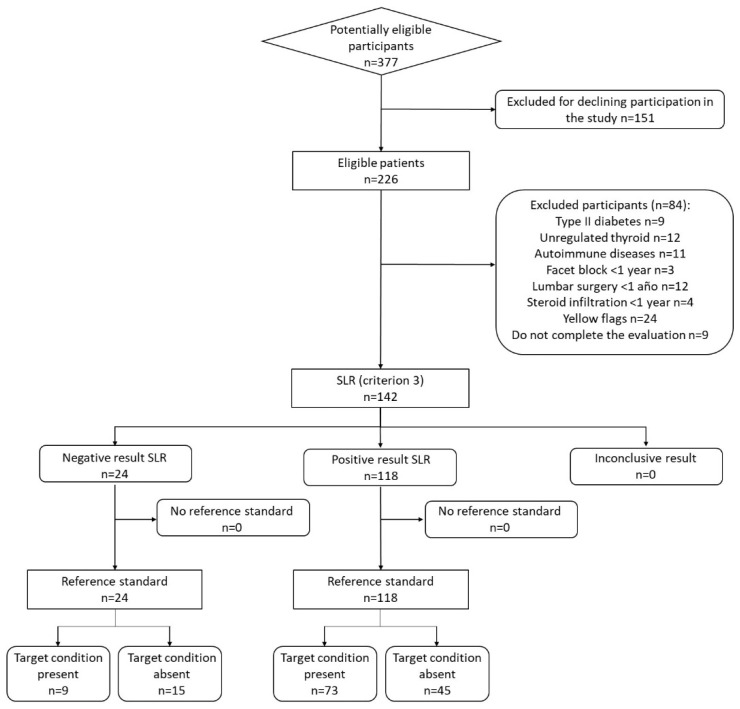
Flow chart of the study profile according to the Standards for Reporting of Diagnostic Accuracy recommendations.

**Figure 2 healthcare-11-03138-f002:**
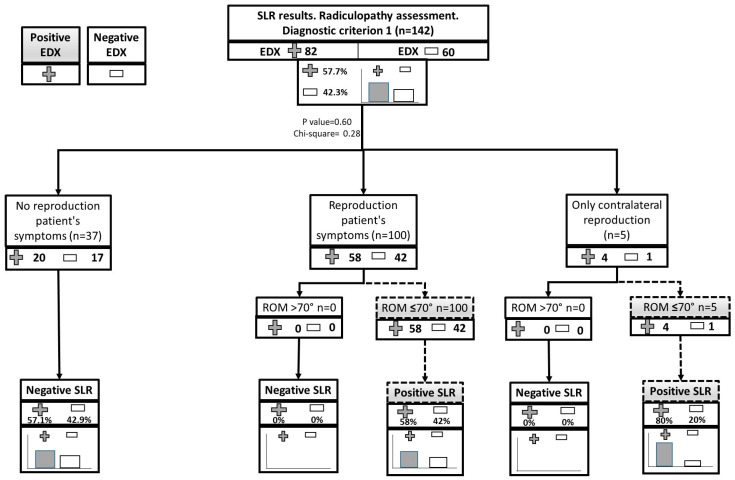
Flow charts of the results of the SLR test for the three diagnostic criteria at each step of the diagnosis of lumbosacral radiculopathy.

**Table 1 healthcare-11-03138-t001:** Descriptive information for participants.

	Participants (*n* = 142)	Radiculopathy (*n* = 82)	Non Radiculopathy (*n* = 60)	*p* Value
Mean age (years)	54.82 ± 12.33	55.53 ± 11.75	53.84 ± 13.12	0.48 *
Gender (female)	57.7%	56.1%	60%	0.64 **
BMI (kg/m^2^)	26.62 ± 4.30	26.78 ± 4.48	26.41 ± 4.08	0.58 *
Dominance (right)	92.3%	93.9%	90%	0.78 **
Symptoms duration (years)	5.44 ± 6.10	6.66 ± 7.26	3.78 ± 3.41	0.03 *
VAS last week	5.31 ± 2.45	5.19 ± 2.47	5.47 ± 2.45	0.51 ***
Oswestry	34.33 ± 18.12	33.24 ± 17.86	35.8 2 ± 18.53	0.41 ***

* Mann–Whitney *u* test; ** chi-square test; *** student-*t* test. Abbreviations: BMI, body mass index; VAS, Visual Analogue Scale.

**Table 2 healthcare-11-03138-t002:** Descriptive information for SLR variables.

SLR Variable		Participants (*n* = 142)	Radiculopathy (*n* = 82)	Non Radiculopathy (*n* = 60)	*p* Value
Neurodynamic responses	Neurodynamic	132 (93%)	77 (93.9%)	55 (91.7%)	0.57 **
	Musculoskeletal	10 (7%)	5 (6.1%)	5 (8.3%)	
Symptom reproduction	Yes	100 (70.4%)	58 (70.7%)	42 (70%)	0.84 **
	No	42 (29.6%)	24 (29.3%)	18 (30%)	
ROM at P1	Affected	39.91° ± 15.21	39.72° ± 16.30	40.16° ± 13.70	0.49 *
	Non-affected	52.62° ± 13.82	53.30° ± 13.83	51.68° ± 13.86	0.73 *
Main symptom	Tension	103 (72.9%)	63 (76.8%)	40 (55%)	0.29 **
	Pain	76 (54.1%)	43 (52.4%)	33 (66.7%)	0.63 **
Symptom location	Calf	20 (14.8%)	14 (17.1%)	6 (10%)	0.27 ***
	Popliteal	21 (14.1%)	12 (14.6%)	9 (15%9	
	Thigh	20 (14.1%)	10 (12.2%)	10 (16.7%)	

* Mann–Whitney *u* test; ** chi-square test; *** Cramer’s-v. Abbreviations: SLR, straight leg raise.

**Table 3 healthcare-11-03138-t003:** Diagnostic accuracy of TND SLR compared to EDX (confidence level: 95%).

	Criterion 1	Criterion 2	Criterion 3
Sensitivity (%)	75.61 (CI 65.71–85.51)	73.17 (CI 62.97–83.37)	89.02 (CI 81.65–96.40)
Specificity (%)	28.33 (CI 16.10–40.57)	33.33 (CI 20.57–46.09)	25.00 (CI 13.21–36.79)
+predictive value (%)	59.05 (CI 49.17–68.93)	60.00 (CI 49.90–70.10)	61.86 (CI 52.68–71.05)
−predictive value (%)	45.95 (CI 28.54–63.36)	47.62 (CI 31.32–63.91)	62.50 (CI 41.05-83.95)
+LR	1.06 (CI 0.86–1.29)	1.10 (CI 0.88–1.37)	1.19 (CI 1.01–1.40)
−LR	0.86 (CI 0.49–1.50)	0.80 (CI 0.49–1.33)	0.44 (CI 0.21–0.94)

Abbreviations: LR, likelihood ratio.

## Data Availability

Any data related to the study can be provided upon a reasonable request.

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
