# Peer review of "Is the Straight Leg Raise Suitable for the Diagnosis of Radiculopathy? Analysis of Diagnostic Accuracy in a Phase III Study"

_healthcare, 2023, doi:10.3390/healthcare11243138_

Round 1
Reviewer 1 Report
Comments and Suggestions for Authors
1. Introduction
lines 46-48 and lines 65-68 seem to convey the same message.
It is advised that the introduction includes the reason EDX are considered the golden standard NDT's can be compared against.
This section needs more information to lead the reader on why this study was necessary. the last paragraph is a brief description of the methodology and should not appear there.
2. Methodology
lines 83-87: it is not clear how the researchers did the power calculation. please provide further information on the tool used and why they chose those values (error, senitivity and specificity).
Line 88: which ethics committee? Hopistal, University, other?
Lines 98-99: which dermatomal and myotomal pattern specifically?
Line 160: the H1 is not described anywhere. perhaps include it in the introduction section if you wish
Results/discussion
the authors wrote that the main symptom of the patients was tightness. this should be discussed and commented at the discussion as this might have influenced the outcome of the study. is tightness the only symptom the authors expected and considered the SLR positive?
Comments on the Quality of English Language
the quality of english is adequate
Author Response
|
Reviewer comments |
Response |
|
lines 46-48 and lines 65-68 seem to convey the same message. |
Thank you very much for your time reviewing the manuscript and for your comments. We agree with the observation that the information can be repetitive. Therefore, the information in the second part of the introduction has been removed. |
|
It is advised that the introduction includes the reason EDX are considered the golden standard NDT's can be compared against. |
Thanks for the comment. This information has been added in the last part of the introduction.
|
|
This section needs more information to lead the reader on why this study was necessary. the last paragraph is a brief description of the methodology and should not appear there. |
Again, thanks for the comment. We have restructured the end of the introduction to include the aspects discussed by the reviewer. We have added a paragraph with the initial hypothesis of the study, which includes the criteria used and reflects the need for the study and we have eliminated this information from the objectives, so that it does not contain such methodological aspects. |
|
2. Methodology lines 83-87: it is not clear how the researchers did the power calculation. please provide further information on the tool used and why they chose those values (error, sensitivity and specificity). |
Thank you very much for the comment. Information regarding power calculation has been increased in the manuscript as follows: “To calculate the sample size, the tables elaborated by Hajian-Tilaki were used [33]. Based on the previous data from the Clinical Neurophysiology Service of the study a prevalence of 50% of the pathology in the study population was taken into account. Based on a confidence interval of 95%, a margin of error of 10% a sensitivity and specificity between 75% and 80% a sample size range of between 123-161 subjects was required”.
|
|
Line 88: which ethics committee? Hospital, University, other? |
It is a local committee that values studies from the entire region (Aragón, Spain) both for universities and hospitals. There is no other specific committee. The specific name of the committee has been added to the manuscript. |
|
Lines 98-99: which dermatomal and myotomal pattern specifically? |
The dermatomal o myotomal pattern of L4, L5 and/or S1. This informacion has been added into the manuscript. |
|
Line 160: the H1 is not described anywhere. perhaps include it in the introduction section if you wish |
We agree with the reviewer. The H1 has been included in the introduction section. |
|
Results/discussion the authors wrote that the main symptom of the patients was tightness. this should be discussed and commented at the discussion as this might have influenced the outcome of the study. is tightness the only symptom the authors expected and considered the SLR positive? |
Thanks for the comment.
Patients could choose between the following categories: pain, stretching, tingling, pricking, numbness, burning or other. This information has been added into the methods section.
|
|
|
|
Reviewer 2 Report
Comments and Suggestions for Authors
I always thought SLR doesn't support disc herniation. Sciatica or nerve tension/entrapment in the lower extremities are more related, I thougt.
Very interesting research and I had fun reading it.
I would like to add few opinions.
Line 108: sample size calculation should be added
Line 122: it seems there is text size defferences
Line 203: population numbers with % would help to interpret the table better.
Line 299: the reccomendation of applying those tests on practice would be great .

Author Response
|
Reviewer comments |
Response |
|
I always thought SLR doesn't support disc herniation. Sciatica or nerve tension/entrapment in the lower extremities are more related, I thougt. |
Thank you very much for taking the time to review the manuscript and for all the comments. We especially appreciate this comment since it is something that the authors share greatly. |
|
Line 108: sample size calculation should be added |
Thank you so much. We agree and the text has been modified to make the sample size calculation clearer. |
|
Line 122: it seems there is text size defferences |
Yes, that’s right. This has been modified, thanks. |
|
Line 203: population numbers with % would help to interpret the table better. |
Thanks, we agree and population numbers have been added in the table. |
|
Line 299: the reccomendation of applying those tests on practice would be great |
Thanks for the comment. The truth is that given the CIs obtained in the study, it seemed too bold to make that statement. That's why we'd prefer not to add it. |
Reviewer 3 Report
Comments and Suggestions for Authors
Dear Authors,
the manuscript is well planned and written. It provides interesting indications for clinical activities. However, please:
1. Improve the readability of Figures, especially Figure 2, or refrain from including them in the manuscript.
With best wishes,
Reviewer.
Author Response
Thank you very much for taking the time to review the manuscript and for all the comments.We appreciate very much your suggestions. The font size has been increased in those parts that were more difficult to visualize.
Reviewer 4 Report
Comments and Suggestions for Authors
1. Line 11: leg-related low back pain is often referred to as leg length discrepancy or leg pain related abnormal pelvic tilting and subsequent low back pain, which may have less relationship with the straight leg raising test. We suggest the authors avoid using this term.
2. Line 17: Briefly describe the three criteria here since abstract should be self-explanatory.
3. Line 19: What does +LR and -LR mean? Using abbreviations without fully spelling them in advance will confuse the readers who only have time for abstract.
4. Line 45: I'm afraid that the value of MRI is not merely diagnosing radiculopathy due to disc herniation. What about spondylolisthesis, trauma-related hematoma formation, discitis, ligamentum flavum hypertrophy, etc.? Can't MRI diagnose these conditions?
5. Line 74: The authors are encouraged to put more emphasis on the importance and the potential impact of this study on clinical practice at the end of this paragraph.
6. Line 83: What's the reference literature that supports the authors to suppose the prevalence of radiculopathy is 50%?
7. Line 86: Describe more about your sample size calculation. What's the effect size to determine the number of participants needed in your study?
8. Line 97: The authors described their inclusion criteria as "intermittent or constant pain in the lumbar area or radiating to a distal extremity, to the gluteal fold". Could it just be myofascial pain in the piriformis and its typical referred pattern of pain? The authors should use "and" and "or" very carefully in describing the inclusion criteria.
9. Line 102: Why are patients with type I DM excluded but not type 2 DM? What's the difference between these two kinds of diseases that involve the peripheral nerves?
10. Line 108: What about motor neuron disease? After all, via electrodiagnostic study, it's difficult to distinguish between polyradiculopathy and motor neuron disease.
11. Line 119: What are the abnormal examinations? Do fibrillations, polyphasic waves, giant waves, and decreased recruitment all count?
12. Line 126: Why was the physical examination performed after the electrodiagnostic study? Shouldn't it be a clinical routine practice to perform the noninvasive tests before the invasive tests?
13. Line 136: What does the word "rectifying" mean? To increase or to decrease?
14. Figure 1: What's the yellow flag in the excluded participants? Although this figure was made according to the Standards for Reporting of Diagnostic Accuracy recommendations, the authors should still describe what's the meaning for reference standard and target condition.
15. Table 1 and 2: Were there any significant differences between patients with radiculopathy and those without?
Author Response
|
Reviewer comments |
Response |
|
1. Line 11: leg-related low back pain is often referred to as leg length discrepancy or leg pain related abnormal pelvic tilting and subsequent low back pain, which may have less relationship with the straight leg raising test. We suggest the authors avoid using this term.
|
Thank you very much for taking the time to review the manuscript and for all the comments. We appreciate this comment and the term has been removed from the manuscript. |
|
2. Line 17: Briefly describe the three criteria here since abstract should be self-explanatory.
|
Thank you very much for the comment. The three criteria have been added in the abstract. |
|
3. Line 19: What does +LR and -LR mean? Using abbreviations without fully spelling them in advance will confuse the readers who only have time for abstract.
|
Abbreviations have been removed and the full name has been used in the introduction section. |
|
4. Line 45: I'm afraid that the value of MRI is not merely diagnosing radiculopathy due to disc herniation. What about spondylolisthesis, trauma-related hematoma formation, discitis, ligamentum flavum hypertrophy, etc.? Can't MRI diagnose these conditions?
|
Thanks for the comment. We fully agree with the reviewer. As lumbar disc herniation is the main mechanical cause of radiculopathy, the information on this aspect had been simplified. However, in agreement with the reviewer, "or other degenerative processes of the lumbosacral complex" has been added.. |
|
5. Line 74: The authors are encouraged to put more emphasis on the importance and the potential impact of this study on clinical practice at the end of this paragraph.
|
Thank you very much for the comment. We agree with the reviewer and the end of the introduction has been modified and more information has been added following the comments of all reviewers. We hope the modifications are helpful and contribute to improve the manuscript. |
|
6. Line 83: What's the reference literature that supports the authors to suppose the prevalence of radiculopathy is 50%?
|
The prevalence value used to calculate the sample size is the actual prevalence of patients diagnosed with lumbosacral radiculopathy who attend the clinical neurophysiology service where the study was carried out. |
|
7. Line 86: Describe more about your sample size calculation. What's the effect size to determine the number of participants needed in your study?
|
Thank you very much for the comment. Information regarding power calculation has been increased in the manuscript as follows: “To calculate the sample size, the tables elaborated by Hajian-Tilaki were used [33]. Based on the previous data from the Clinical Neurophysiology Service of the study a prevalence of 50% of the pathology in the study population was taken into account. Based on a confidence interval of 95%, a margin of error of 10% a sensitivity and specificity between 75% and 80% a sample size range of between 123-161 subjects was required”. |
|
8. Line 97: The authors described their inclusion criteria as "intermittent or constant pain in the lumbar area or radiating to a distal extremity, to the gluteal fold". Could it just be myofascial pain in the piriformis and its typical referred pattern of pain? The authors should use "and" and "or" very carefully in describing the inclusion criteria.
|
Thanks, we have changed “or” by “and” to clarify this raised aspect. |
|
9. Line 102: Why are patients with type I DM excluded but not type 2 DM? What's the difference between these two kinds of diseases that involve the peripheral nerves?
|
Thanks for the comment. We understand that type I diabetes is a systemic disease that often causes other health problems. While type II diabetes is more related to lifestyle habits and does not usually have as many associated pathologies. |
|
10. Line 108: What about motor neuron disease? After all, via electrodiagnostic study, it's difficult to distinguish between polyradiculopathy and motor neuron disease.
|
As we understand, Neurophysiologists evaluated both, using ENG and EMG, in such a way that they use ENG to rule out other types of neuropathies other than lumbar radiculopathy. |
|
11. Line 119: What are the abnormal examinations? Do fibrillations, polyphasic waves, giant waves, and decreased recruitment all count?
|
Positive EMG: Pathological findings included (Levin 2002;Gutiérrez-Rivas 2007):- When inserting the concentric needle electrode. Presence of increased “insertion activity”.- At rest. Presence of “fibrillation potentials”, “positive waves”, “fasciculations” and/or “complex repetitive discharges/high frequency discharges”.- In voluntary contraction. Presence of neuropathic motor unit potentials: “increased amplitude”, “increased duration” and/or “increased polyphasia index”.- In maximum contraction. Abnormal recruitment of the motor unit: “recruitment pattern”. - Absence of H reflex for evaluation of the S1 root. |
|
12. Line 126: Why was the physical examination performed after the electrodiagnostic study? Shouldn't it be a clinical routine practice to perform the noninvasive tests before the invasive tests?
|
In this case, the sample was taken from patients who were scheduled to undergo the EDX. In order not to delay the service, the patient first came to their appointment and then we took them. Furthermore, we only took those that had been specifically EMG done and we can only know that after the fact. |
|
13. Line 136: What does the word "rectifying" mean? To increase or to decrease?
|
We completely agree that the word rectifying is confusing. For this reason, it has been replaced by the word reducing. Thanks. |
|
14. Figure 1: What's the yellow flag in the excluded participants? Although this figure was made according to the Standards for Reporting of Diagnostic Accuracy recommendations, the authors should still describe what's the meaning for reference standard and target condition.
|
Patients were excluded because they showed clear behaviors of psychological alteration. For example, in one case, the patient spoke of aliens who wanted to poison humans. In another, the patient referred to having undergone skull surgery. In addition to not presenting any visible signs of said intervention, there was no information about it in his medical history.As the SLR response classification requires clear communication with the patient and a sometimes detailed description of the symptoms, it was decided not to include any of them in the study. |
|
15. Table 1 and 2: Were there any significant differences between patients with radiculopathy and those without?
|
From Table 1, only the variable “duration of symptoms” had a significant comparison.From Table 2, no variable had a significant comparison. Information regarding significance has been added into the manuscript. |
Reviewer 5 Report
Comments and Suggestions for Authors|
|
2. Materials and Methods
“Inclusion criteria were: patients aged between 18 and 95 80 years”maybe it is more correct to indicate up to 18 years old?
“An approximate prevalence of 50% of the pathology in 83 the study population was taken into account..” can you explain it? I can not understand
“Local ethics committee ap-88 proved the protocol of this study (PI21/073).” Can you explain wich local ethics committee was?
Author Response
|
Reviewer comments |
Response |
|
“The research titled “Is the straight Leg Raise Suitable for the Diagnosis of Radiculopathy? Analysis of Diagnostic Accuracy in a Phase III Study” it is a very novelty topic and pertinent. The quality of the research is quite good, but maybe there are some comments that can improve the manuscript.
|
Thank you very much for taking the time to review the manuscript and for all the comments. |
|
The authors affiliation superindice is missing, the author guideline is: Firstname Lastname 1, Firstname Lastname 2 and Firstname Lastname 2,* |
The affiliation superindice has been added. |
|
2. Materials and Methods
“Inclusion criteria were: patients aged between 18 and 95 80 years”maybe it is more correct to indicate up to 18 years old?
|
We agree and the sentence has been modified as suggested by the reviewer. Thanks. |
|
“An approximate prevalence of 50% of the pathology in 83 the study population was taken into account..” can you explain it? I can not understand |
The prevalence value used to calculate the sample size is the actual prevalence of patients diagnosed with lumbosacral radiculopathy who attend the clinical neurophysiology service where the study was carried out. |
|
“Local ethics committee ap-88 proved the protocol of this study (PI21/073).” Can you explain which local ethics committee was?
|
Yes, the name of the ethics committee has been added in the manuscript. |
Round 2
Reviewer 2 Report
Comments and Suggestions for Authors
I again enjoyed the writing, and everything looks great except one last thing.
I would like to see the recommendation how to apply SLR with radiculopathy when practicing on the field.
other than the specific practical recommendation, the it would make a great article.
Thank you.
Author Response
Thank you very much again for your time and effort in reviewing the manuscritp.
We appreciate very much your comment and the following information has been added in the manuscript.
"Specifically, the clinical interpretation of the results obtained would be that, if the SLR test does not reproduce the patient’s symptoms and there are no asymmetries in the ROM or symptoms location even though the structural differentiation has produced a change in the symptoms, the probability that the patient does not have lumbosacral radiculopathy is high."